# Clinical Validation of Digital Healthcare Solutions: State of the Art, Challenges and Opportunities

**DOI:** 10.3390/healthcare12111057

**Published:** 2024-05-22

**Authors:** Mar Gomis-Pastor, Jesús Berdún, Alicia Borrás-Santos, Anna De Dios López, Beatriz Fernández-Montells Rama, Óscar García-Esquirol, Mònica Gratacòs, Gerardo D. Ontiveros Rodríguez, Rebeca Pelegrín Cruz, Jordi Real, Jordi Bachs i Ferrer, Adrià Comella

**Affiliations:** 1Digital Health Validation Center, Hospital de la Santa Creu i Sant Pau, Sant Pau Campus Salut Barcelona, 08041 Barcelona, Spain; jberdunp@santpau.cat (J.B.); aborrass@santpau.cat (A.B.-S.); adedios@santpau.cat (A.D.D.L.); bfernandezm@santpau.cat (B.F.-M.R.); gontiveros@santpau.cat (G.D.O.R.); rpelegrin@santpau.cat (R.P.C.); jrealg@santpau.cat (J.R.); acomella@santpau.cat (A.C.); 2Institut de Recerca Sant Pau (IR SANT PAU), Sant Quintí 77 79, 08041 Barcelona, Spain; 3Pharmacy Department, Hospital de la Santa Creu i Sant Pau, IIB Sant Pau, 08041 Barcelona, Spain; 4Barcelona Health Hub, 08025 Barcelona, Spain; oges@mediktor.com; 5DAP-Cat Group, Unitat de Suport a la Recerca Barcelona, Fundació Institut Universitari Per a la Recerca a l’Atenció Primària de Salut Jordi Gol i Gurina (IDIAPJGol), 08028 Barcelona, Spain; monica.gratacos@gmail.com; 6Departament d’Economia i Organització d’Empreses, Universitat de Barcelona (UB), 08036 Barcelona, Spain; jbachs@ub.edu

**Keywords:** digital health technologies, clinical validation, adoption challenges, evaluation frameworks, empirical evidence, technology transfer, eHealth, healthcare digitalisation, innovation, patient care

## Abstract

Digital health technologies (DHTs) at the intersection of health, medical informatics, and business aim to enhance patient care through personalised digital approaches. Ensuring the efficacy and reliability of these innovations demands rigorous clinical validation. A PubMed literature review (January 2006 to July 2023) identified 1250 papers, highlighting growing academic interest. A focused narrative review (January 2018 to July 2023) delved into challenges, highlighting issues such as diverse regulatory landscapes, adoption issues in complex healthcare systems, and a plethora of evaluation frameworks lacking pragmatic guidance. Existing frameworks often omit crucial criteria, neglect empirical evidence, and clinical effectiveness is rarely included as a criterion for DHT quality. The paper underscores the urgency of addressing challenges in accreditation, adoption, business models, and integration to safeguard the quality, efficacy, and safety of DHTs. A pivotal illustration of collaborative efforts to address these challenges is exemplified by the Digital Health Validation Center, dedicated to generating clinical evidence of innovative healthcare technologies and facilitating seamless technology transfer. In conclusion, it is necessary to harmonise evaluation approaches and frameworks, improve regulatory clarity, and commit to collaboration to integrate rigorous clinical validation and empirical evidence throughout the DHT life cycle.

## 1. Introduction

Digital health technologies (DHTs) constitute an emerging field at the intersection of health, medical informatics, and business, with the primary goal of improving the quality of patient care through personalised and patient-centric digital approaches [1,2]. The National Institute for Health and Care Excellence (NICE) defines digital products as those intended to benefit people or the wider health and social care system, including smartphone applications (mobile health [mHealth]), standalone software, online tools for treating or diagnosing conditions, preventing ill health or improving system efficiencies, and programmes that can be used to analyse data from medical devices such as scanners, sensors or monitors [3]. Moreover, the World Health Organization (WHO) defines digital health as knowledge and practice associated with the development and use of digital technologies to improve health, encompassing areas such as artificial intelligence (AI), big data, blockchain, health data, health information systems, the infodemic, the Internet of Things, interoperability, and telemedicine [4].

Over the past few decades, the rapid surge in digital health innovations has propelled the field into continuous and exponential growth, fundamentally transforming the healthcare landscape [5]. This remarkable growth can be largely attributed to the increase in government initiatives promoting the adoption of eHealth solutions [6]. However, the process of digitising or adopting digital technologies within the healthcare sector has been slower compared to other sectors, a fact often attributed to the complex interplay of existing regulatory requirements, the existence of multiple funders or payors in a country or region that complicate the purchase of DHTs, the critical nature of healthcare decisions, the need to handle personal health data with the utmost care, and variations in digital competences among health professionals [7,8,9,10]. Furthermore, some professionals exhibit a resistance to change, and there is a shortage of resources, particularly in personnel, within information systems and healthcare departments, hindering the full adoption of the technology. Indeed, the digital transformation of patient care requires a cultural change and involves patient–provider interactions, institution-to-institution data sharing, and peer-to-peer communication, thus requiring challenging collaborative efforts among healthcare providers, technology developers, regulatory bodies, and policymakers [2].

As digital technologies rapidly advance to integrate into healthcare systems, there is a growing demand for robust clinical validation to ensure the efficacy and reliability of these innovations [11]. However, stakeholders face challenges in comprehensively understanding the clinical robustness and claims made by the tech industry. This is evidenced by the relatively low level of clinical robustness exhibited by many digital health companies, with limited regulatory filings, clinical trials, and publicly available data [12]. This underscores a crucial gap that suggests the potential for enhanced clinical validation efforts, a facet that could prove mutually beneficial for both companies and customers [13]. However, there is a lack of clarity regarding the requisite procedures for obtaining clinical evidence. This ambiguity extends not only to the terminology employed in the context of clinical validation but also to the identification of optimal methodologies for conducting studies, determining the minimum validation requirements necessary for product marketing, and facing the challenges associated with formulating regulations amid the rapid growth and change in the technologies under development. This lack of a well-defined roadmap poses a considerable challenge, ultimately hindering the implementation of effective and meaningful clinical validation strategies by digital health developers.

Our study has two main objectives. First, we aim to provide readers with a comprehensive understanding of current trends in the dynamic landscape of DHTs. Second, we aim to provide a better understanding of the complex challenges, methodologies, and frameworks that digital health developers face. By identifying the main obstacles and exposing the methodologies and frameworks that support digital health development, our study aims to equip stakeholders with actionable knowledge and strategic insights. Employing a comprehensive literature search, we conducted a bibliometric analysis until the 23 April 2024 to evaluate historical and ongoing research trends in the field of DHTs. Additionally, for a comprehensive understanding of the primary challenges, current methodologies, and frameworks embraced by digital health developers, we conducted a narrative review focusing on peer-reviewed publications from the 1 January 2018 to the 23 April 2024, addressing either the conceptual exploration of eHealth interventions or discussions on the clinical validation, quality, and accreditation of DHTs. Finally, we introduce the key components of the Digital Health Validation Center, an initiative aimed at streamlining the process of generating robust clinical evidence for technology companies and healthcare providers, ultimately fostering the adoption of digital technologies into clinical practice.

## 2. Materials and Methods

### 2.1. Search Strategy

In the process of article selection for the literature search, we adhered to the “Preferred Reporting Items for Systematic Reviews and Meta-Analysis” (PRISMA) methodological approach [14]. A health science-specialised librarian conducted a Boolean search in the PubMed medical database on the 23 April 2024. The following operators were used: (“digital health” [Other Term]) OR (“Digital health” [Title/Abstract]) OR (eHealth [Other Term]) OR (mHealth [Other Term]) OR (Telemedicine [MeSH Major Topic]) AND (“Clinical validation” [Other Term]) OR “clinical validation” OR “clinical quality” OR “App quality” OR “accreditation” OR (“Validation” [Title/Abstract]).

### 2.2. Bibliometric and Visual Analysis

To assess the evolution and current state of research in the field of DHTs, we conducted a bibliometric and visual analysis of all retrieved articles to identify key trends, patterns, and interrelationships. The data required for the bibliometric analysis were obtained from the NCBI Entrez API, which allows a computer program to replicate bibliographic searches in PubMed. This process yields a dataset containing structured metadata extracted from the retrieved articles. The following bibliometric indicators were extracted from each paper and imported for in-depth analyses: publication year, journal name, journal country, abstract, and keywords. Finally, impact factors, quartiles, and journal categories were obtained from the Journal Citation Reports (Clarivate).

A descriptive analysis was conducted to examine the frequencies of these variables. Additionally, we performed graphical analyses, generated a word cloud illustrating the distribution of keywords and mesh terms, and assessed frequency patterns based on the region and country of the first author. We identified variables related to the study design and evaluation domains through word searches in the title, abstract, and keywords. Appendix A provides a list of synonymous words used for design and domain categorisation. To explore the relationship between study design (observational/experimental) and domains, we constructed a descriptive bivariate table. The data management and analysis were conducted using the R Core Team (version 4.3.0) software environment [15], the bivariate tables using the compareGroups R package (version 4.6.0) [16,17], and the graphical analyses using the ggplot2 (version 3.5.0), maps (version 3.4.1), and wordcloud2 (version 0.2.1) R packages [18,19,20]. The analysis scripts are outlined at https://github.com/CVCSD/Clinical-Validation-StateOfArt (accessed on 10 May 2024).

### 2.3. Literature Review and Data Collection

This approach was chosen to capture a comprehensive overview of the evolving landscape of DHTs, encompassing both theoretical and practical dimensions. To ensure a focus on the most recent trends in the dynamic field of DHTs, the narrative review only considered papers published from the 1 January 2018 to the 23 April 2024.

For study selection, all identified articles in English were exported to a reference manager software (Mendeley Desktop (version 1.19.5), Elsevier, Amsterdam, The Netherlands). Two authors independently assessed the titles and abstracts of the identified articles, focusing on those that included either a conceptual exploration of eHealth interventions or discussions on the clinical validation, quality, and accreditation of DHTs. Articles centred around specific pathologies were excluded from consideration. This decision was made to maintain a broad scope and prioritise discussions and insights relevant to the broader field of DHTs. While pathologies represent important areas of research and innovation, our study sought to provide a holistic perspective on DHTs that transcends individual disease states.

Articles identified as relevant were retrieved in full text and subsequently reviewed by the same authors for inclusion, adhering to predefined eligibility criteria. Data extraction from the selected articles was conducted using a standardised form in Microsoft Excel, encompassing articles addressing challenges encountered during the evaluation of DHTs and those proposing frameworks for DHT evaluation. For articles presenting multidimensional frameworks, the extracted fields included the type of DHT, scope (development, design, or evaluation), target population (decision makers, governments, clinicians, health researchers, or users), developers (technology companies, developers, researchers, eHealth experts, practitioners, or final users), and dimensions/criteria/steps for assessment.

## 3. Results

The initial literature search identified 1421 publications used to assess historical and ongoing trends in research status in the field of DHTs (Figure 1). From the initially selected pool of peer-reviewed publications, we identified 1053 papers published between January 2018 and April 2024 (Figure 1). Out of these, 41 (3.9%) met the eligibility criteria for full-text review. After the in-depth examination of the complete texts, seven papers were excluded as they focused on validating a specific framework within the context of a particular disease. Furthermore, after reviewing the references included in the selected articles, 20 additional papers were incorporated into the qualitative synthesis.

### 3.1. Bibliometric and Visual Analysis

The annual publication output followed a sustained and exponential growth since 2014, accompanied by a gradual rise in prevalence within journals classified in quartiles one or two (Figure 2a). Concerning the scope of journals featuring publications in the field of digital health, 21.5% of the articles were in the “health care sciences” category, 18.6% in journals related to “medical informatics”, and 9.3% in “public, environmental, and occupational health” (Figure 2b). All other categories collectively constituted less than 4% of the publications. The analysis of prolific authors’ geographical distribution showed that Europe was the predominant contributing region, accounting for 33.9% of the publications, followed closely by the USA (22.9%; Figure 3a).

To identify key themes and topics prevalent in the literature, we conducted a word cloud technique to obtain a visual representation where words are sized according to their frequency of occurrence in the text. From this analysis, we observed that terms such as “eHealth”, “mobile health”, “mobile phone”, and “validation” emerged as the more prominent keywords (Figure 3b).

The descriptive analysis of the studies showed that 54.0% employed an observational approach, while 46.0% utilised an experimental methodology. In terms of study design, clinical trials accounted for 41.3% of the studies, followed closely by cross-sectional studies at 36.6%. Less common were cohort studies (11.6%), systematic reviews (10.2%), and case–control studies (0.38%).

When analysing the papers based on evaluation domains, we identified eight distinct clusters: usability/satisfaction (mentioned in 18.4% of papers), security/privacy (13.8%), feasibility/proof of concept (10.6%), functionality (7.0%), effectiveness (9.6%), efficacy (5.8%), efficiency/cost–utility (7.5%), and implementation (6.5%). Across both experimental and observational designs, all dimensions were uniformly assessed, except for feasibility (proof of concept), effectiveness, and efficacy dimensions. These three dimensions were more frequently evaluated in experimental study types compared to observational studies (*p* < 0.05).

### 3.2. Literature Review

In total, we examined 54 articles, categorising them into two main groups and subcategories, as illustrated in Appendix A. The first category included articles addressing challenges encountered in the evaluation of DHTs. The second category encompassed articles discussing frameworks for DHT evaluation. We defined the framework as a structured guideline directing attention to specific attributes of the DHT considered most relevant for evaluation purposes [21]. This section was further subdivided into reviews of published frameworks for evaluation, articles proposing multidimensional frameworks, and articles focused on specific evaluation dimensions.

#### 3.2.1. Challenges Faced When Evaluating Digital Health Technologies

Articles addressing challenges in DHT evaluation were categorised as follows (Appendix A): (1) offering a general and conceptual perspective [22,23,24,25]; (2) exploring the adoption and implementation of these technologies within clinical practice or complex healthcare systems [26,27,28,29,30,31,32]; (3) assessing accreditation processes [33,34]; or (4) delving into ethical, legal, and social implications [32,34,35,36,37].

##### Challenges from a General and Conceptual Perspective

In this section, we present an overview of the opinions and viewpoints expressed in four articles that provide a general and conceptual perspective on challenges in DHT evaluation [22,23,24,25].

One of the key challenges discussed in the literature pertains to accurately defining DHT evaluation. One article emphasises that, despite a plethora of terminologies, DHT evaluation can be categorised into verification, analytical validation, and clinical validation, which must demonstrate clinical effectiveness, efficacy, and safety [23]. The authors emphasise that the necessary level of clinical validation depends on the complexity of these technologies [23,25]. Indeed, while simpler applications may need less evidence, the validation process for complex diagnostic algorithms or medical devices demands thoroughness to secure regulatory approval [23].

Authors also highlight the challenges associated with adapting traditional evaluation models, such as Health Technology Assessments (HTAs), for DHTs. While HTAs were originally designed for pharmaceuticals, medical devices, and procedures, they may not cover all relevant dimensions, such as usability/satisfaction, accessibility, and data protection, particularly for emerging technologies like mHealth, AI, and robotics [22,23]. Moreover, pharmaceutical models overlook the unique characteristics of complex eHealth interventions due to user-dependent effectiveness, variations among similar apps, and fluctuating cost structures [24,25].

Another perspective discussed in the literature is the practicality of existing evaluation methods, particularly in the context of DHTs. Existing generic evaluation models use extensive double-blind multicentre randomised controlled trials (RCTs), which are often impractical for DHTs [23,24,25]. Actually, only a few validation frameworks have undergone peer review or regulatory recognition [23].

Authors also address deployment strategies for DHTs and their potential impact on patient safety. Many digital health products adopt the “implement now, validate later” strategy, which may expedite the deployment of DHTs but carries significant risks that can impact patient safety [23]. Consequently, establishing consensus on evidence levels and maintaining rigorous clinical research throughout a product’s lifecycle emerges as a pivotal challenge in the field [23].

##### Challenges in Adoption and Implementation

Seven articles examined adoption or implementation challenges related to the integration of DHTs into clinical practice or complex healthcare systems [26,27,28,29,30,31,32]. A study analysed the implementation of DHTs in 17 integrated chronic care programs across eight European countries participating in the “Sustainable intEgrated care modeLs for multimorbidity: delivery, Financing, and performancE (SELFIE)” Horizon 2020 project. The results revealed partial electronic health records implementation, with limited support for telemonitoring services and patient self-management through health information exchange platforms, and barriers like funding constraints, limited research capacity, data security concerns, and potential data misuse [26].

The limited integration of DHTs into clinical care stems from challenges such as the inadequate dissemination of appropriate interventions among both patients and healthcare providers [27]. This situation arises because, despite the multitude of available apps, each is characterised by its unique functionality, impact, and cost, coupled with limited evidence of effectiveness [27,28,29]. Indeed, the development and implementation of health apps often deviate from standards due to suboptimal design, inadequate testing, and a lack of comprehensive data and long-term outcome assessments, resulting in not guaranteed issue-free health apps [31]. To enhance integration, essential steps involve establishing the best practices, implementing robust validation approaches, enhancing education and awareness for clinicians, and providing user support and guidance [27,28,32]. Additionally, considerations for adoption and integration into complex health systems extend beyond clinical value, encompassing factors like the development source, potential financial value, and data access and interoperability [29,32]. Finally, it is important to implement rigorous procedures to assess the safety of new DHTs, and it is essential to incorporate perspectives from low- and middle-income countries (LMIC) where the use of digital tools is on the rise [31].

##### Challenges of Accreditation Processes

The regulatory landscape for DHTs is dynamic and evolving, responding to technological advancements and emerging challenges, with assessment frameworks across jurisdictions serving different purposes, such as regulating market access, coverage, or reimbursement [25,38].

A structured survey conducted in 2018 across 30 European countries revealed diversity in accreditation mechanisms for websites and mobile applications, with only six countries having established evaluation protocols for mHealth apps (Estonia, Germany, Portugal, Slovenia, Spain, and the UK) [33]. Health website accreditation was rare, observed in only eight countries (Austria, Croatia, Estonia, France, Germany, Portugal, Spain, and the UK), some of which used the Health On the Net (HON) code for certification [33].

The 2022 Survey on Digital Health in the WHO European Region offered an overview of the current state of digital health initiatives [34]. With a focus on policies and strategies, the survey revealed that nearly all EU countries indicated the possession of a national digital health policy or strategy, with the exception of Armenia, Bulgaria, Luxembourg, Monaco, North Macedonia, the Republic of Moldova, Slovakia, and Slovenia. Furthermore, only 19 members (36%) reported having established guidance for the evaluation of DHTs, while others either lacked such guidance, were unaware of its existence, or did not provide a response.

Two recent articles summarised guidance and regulatory frameworks for digital health innovators from international organisations, including the US Food and Drug Administration (FDA), which has established guidelines for approving and clearing all DHTs, distinguishing between Software as a Medical Device (SaMD) and Software in a Medical Device (SiMD) [25,38]. The WHO provides practical guidance applicable to all DHTs, while the UK’s NICE evidence standards framework for DHTs employs a risk-based approach based on digital tool functionality, considered a mature evaluation model [25,38]. The Continua Design Guidelines (CDG) from the Personalized Connected Healthcare Alliance (PCHA) apply to connected devices and mobile applications [25,38]. In the European Union, DHTs are subject to regulatory requirements for medical devices, involving conformity assessment through the CE marking process to ensure safety and performance standards are met [25,38]. Within Europe, both France, represented by the Haute Authorité de Santé (HAS), and Germany, through the Federal Institute for Drugs and Medical Devices (BfArM), employ risk-based approaches and have reference documents to assess coverage and reimbursement decisions [25]. In France, the focus lies on health apps and smart devices, while in Germany, the scope extends to any digital health application as outlined in the German Digital Health Application (DiGA) Guide [25].

##### Challenges in Ethical, Legal, and Social Implications

We identified one literature review addressing the ethical, legal, and social issues (ELSI) arising from DHTs and health data processing, including telehealth, telemedicine, and AI and machine learning [35]. The article identified ethical concerns within this context encompassing issues like the promotion of patient autonomy and empowerment; the design, obtainment, and interpretation of informed consent; strengthening the doctor–patient relationships; identity confirmation and authentication; defining professional duties and responsibilities; ensuring fair risk, benefit, and costs distribution; maintaining confidentiality, and addressing the potential dehumanisation of care [35]. Legal considerations were considered to involve aspects such as data protection rights; data access; the return of information; non-discrimination; data ownership rights; establishing fair, transparent, and harmonised data sharing rules; managing the return of information; preventing discrimination; and addressing jurisdiction and licensure issues for telemedicine [35]. Lastly, social challenges mentioned included considering variations in digital literacy levels among patients and health professionals; issues of inequality and social stigma; or the impact on health access, considering economic, geographical, and informational factors [35].

A recent systematic review examined the multifaceted challenges, including legal and ethical barriers, to eHealth implementation in Europe [32]. The review emphasised the critical importance of safeguarding data confidentiality and ensuring secure data management. Ethical considerations such as consent, responsibility, and liability were identified as key barriers, requiring a nuanced approach due to their interconnected nature. Moreover, the validation of eHealth technologies was considered to be hindered by limited clinical and socio-economic evidence, emphasising the necessity for robust ethical frameworks aligned with legal constructs to guide the development and implementation of eHealth solutions effectively.

Concerning the utilisation of big data in the healthcare domain, findings from the 2022 Survey on Digital Health in the WHO European Region [34] revealed that government policies or strategies oriented towards this purpose, encompassing its application in both the public and private sectors or the extraction of data from Electronic Health Record (EHR) systems, were adopted in only 17 countries (32%). An article discussing the use of blockchain technology in health data sharing compared adherence to personal data protection and healthcare regulatory guidelines across different countries, namely Europe, the US, Canada, Australia, Japan, and Brazil [36]. While each regulation targets data privacy and security concerns, significant differences exist in their scope, consent procedures, transparency, accountability, and enforcement mechanisms. Nevertheless, key principles like informed consent, transparency, accountability, and impact assessments were commonly emphasised across regulations [36]. Similarly, a systematic review examining the integration of AI with digital health data underscored the importance of transparency and accountability to foster trust among clinicians and patients in this technology [37].

#### 3.2.2. Reviews of DHTs Evaluation Frameworks

We identified nine reviews of existing DHT evaluation frameworks [21,22,39,40,41,42,43,44,45,46].

Overall, the landscape of DHT evaluation is characterised by conceptual frameworks lacking practical guidance [41]. Moreover, almost half of new mHealth app publishers are from Europe, and challenges persist in adapting evaluation methodologies for LMICs [46].

Regarding the framework’s components, there are two main approaches for evaluating eHealth interventions [40]: one examines different evaluation aspects (domains), which can be either continuous and iterative or step-wise. The other follows a product lifecycle perspective, where specific aspects of evaluation vary depending on the phase, ranging from pre-market development to post-market or post-intervention performance [40]. However, these two approaches do not interact with each other, resulting in uncertainty about which aspects to assess during each phase of evaluation [40].

In addition to the diversity in evaluation approaches, all reviews highlight a significant variability in evaluation domains, quality criteria, and definitions for each criterion used for evaluation [21,39,41,42,45,46]. This variability is partially attributed to the lack of regulatory clarity and the absence of institutionalised quality controls in many countries [41,45]. Consequently, developers often create assessment domains, scales, or questionnaires themselves, termed “boutique” criteria [21,40,41,44], raising concerns about standardisation and generalizability [39,40,41,44].

Another frequently cited challenge is that some frameworks fail to consider crucial assessment criteria, leading to incomplete evaluations focused on specific issues [41]. For instance, while most studies assess the quality of health apps, only approximately half of them evaluate health information, and clinical effectiveness supported by empirical evidence is rarely included as a criterion for app quality [21,41,46], significantly diminishing the potential societal impact of these tools [41]. Furthermore, frameworks often neglect factors like app engagement prediction and app customizability [42].

As highlighted in conceptual articles, the practical adaptability of the current framework’s HTAs is limited, as they are primarily designed for policy decision-making [22,23]. A systematic review of the suitability of evaluation frameworks for use in the HTAs of mobile medical applications found that, for therapeutic tools, none of the frameworks addressed the key elements to demonstrate effectiveness, and only 16% considered the quality of the evidence base [43]. Moreover, reviewed frameworks did not account for the potential direct costs or impacts of the app on the healthcare system.

#### 3.2.3. Articles Proposing a Multidimensional Evaluation Framework or Model

We identified 15 articles proposing a single multidimensional framework/model to evaluate DHTs, 6 of them already included in the aforementioned reviews of existing frameworks. In Appendix A, we provide a list with details on each of the articles and frameworks examined, offering an overview of the field’s status from 2018 to the time of compilation. Overall, the assessed frameworks were hardly comparable because of the substantial variations in the domains and quality criteria they assessed to evaluate DHTs.

Out of the 15 examined articles, the focus was primarily on evaluating mHealth apps, with nine frameworks (60%) specifically tailored for this purpose [45,47,48,49,50,51,52,53,54]. Four frameworks were applicable to all eHealth technologies [11,51,55,56], one extended its scope to cover both mHealth apps, AI, and robotics [22], and another one proposed quality criteria for AI-based prediction models [57].

A total of 12 of the 14 frameworks (87%) were developed relying on eHealth experts, healthcare professionals or researchers’ opinions [11,45,47,48,49,50,51,53,54,55,56,58]; additionally, some frameworks incorporated input from other stakeholders, such as technology companies, family members, policymakers, potential users, or app developers and engineers [22,49,52,53,58,59].

The target audience for the assessed frameworks varied. Six focused on all stakeholders, including governments, patients, clinicians, and health researchers [45,47,53,54,59,60]. Three targeted both patients and clinicians [49,50,56], two targeted decision-makers [22,52], two targeted only healthcare professionals [48,51], and one specifically targeted eHealth implementers in developing countries [55].

Seven frameworks emerged from a literature review of existing models [45,48,49,53,55,56,59]. Two were built based on the evaluation of existing accredited apps [47,50], while two proposed modifications to the evaluation domains of traditional HTAs [22,52]. One study proposed a framework based on evaluation areas assessed by representative resources like the NHS app library [11], another created a list of criteria based on expert consensus to develop a digital health platform [51], and another proposed a modification of an existing evaluation tool [54].

More than half of the frameworks (*n* = 9; 60%) were developed as scales or guides to assess or rank the app quality to support decision-making in choosing mHealth apps [45,47,48,49,50,51,53,54,59]. The remaining frameworks proposed more general guides for developing and evaluating DHTs [11,22,52,55,56]. Among the five general models, two followed the lifecycle approach [11,56], two adopted the HTAs model [22,52], and one presented a generic model distinguishing between monitoring and evaluation dimensions with performance indicators [55].

#### 3.2.4. Articles Proposing or Reviewing Specific Evaluation Dimensions

##### Safety

Two scoping literature reviews examined safety and privacy concerns in consumer-facing mHealth apps [61,62]. In the first review [61], 84% of safety issues were related to content quality, including (1) incorrect or incomplete information presented by the app; (2) variations in content quality among apps addressing similar domains; as well as (3) inaccuracies in the output of apps offering calculations and diagnostic results, coupled with inappropriate responses to consumer needs. Software functionality concerns (16%) involved feature gaps, delayed processing, faulty alarms, and unresponsiveness to safety-critical information.

The second review, addressing safety and privacy techniques and frameworks proposed for mHealth apps [62], found that a third of included papers approached privacy and security as a combined facet within a general evaluation of mHealth design and only 40% of the frameworks specifically targeted the evaluation of security and/or privacy. Furthermore, only a few studies (11.4%) mentioned compliance with existing regulations in their assessments and design recommendations. The authors emphasised the distinction between security and privacy, highlighting that security involves safeguarding data from unauthorised access, whereas privacy encompasses an individual’s right to maintain control over their private data and communications. Consequently, an exclusive focus on security may inadvertently heighten surveillance and data collection, consequently introducing potential privacy risks.

##### Economic Evaluation

We identified one systematic review that focused on the quality of the economic evaluation of various eHealth technologies, including computer decision support systems, web-based physical activity interventions, internet-delivered cognitive behavioural therapies, telecare, and telehealth [63].

The review revealed that 82% of the studies used statistical regression modelling, 18% used simulation modelling, and one study employed both methods to predict costs and quality-adjusted life years (QALYs). However, the overall findings indicated a lack of convincing evidence to determine whether the use of eHealth technologies for delivering healthcare to older adults would demonstrate value at any acceptable level of investment. For example, many studies predominantly adopted one-year horizons, lacked an adequate representation of the broader societal perspective, provided unclear descriptions of healthcare resource use and cost components, presented ambiguous explanations of the methods used to estimate model parameters in model-based studies, and showed insufficient reporting on uncertainty analysis, whether deterministic or probabilistic.

##### Evidence of Clinical Effectiveness

Overall, we reviewed nine articles that assessed the evaluation of the clinical effectiveness of DHTs. Two of the articles specifically evaluated the alignment of medical claims made by digital companies and mHealth apps with robust clinical evidence [12,64]. The remaining seven articles focused on methodological approaches used in gathering clinical evidence [38,65,66,67,68,69,70].

In a cross-sectional observational analysis, researchers examined the substantiation of claims made by digital health start-ups on their websites [12]. The evaluation employed a clinical robustness score, defined as the sum of regulatory filings and completed clinical trials by each company, serving as proxies for effectiveness. The average clinical robustness score across all companies was notably low, with 44% of companies scoring 0 and only 20% achieving a score of at least 5, indicative of rigorous testing. Notably, diagnosis companies had the highest average clinical robustness scores, followed by treatment companies and prevention companies. Another study examined the medical claims and scientific substantiation of 120 health apps available on Apple iTunes and US Android Google [64]. Although none of the apps had FDA marketing approval to make explicit medical claims, almost half presented claims that could potentially be interpreted as medical. Furthermore, scientific evidence to support these claims was nearly absent, raising concerns about their validity and effectiveness for their users.

From the conceptual point of view, although RCTs are widely acknowledged as the gold standard for assessing healthcare interventions, the rapid evolution of digital products has limited their applicability, leading to the prevalent use of prospective studies, particularly pre-post designs [38]. Despite their utility, these designs are deemed suboptimal for evaluating medium to long-term clinical outcomes, creating a gap between cost-effective early stage approaches and the higher-cost methods necessary for broader acceptance [38]. As an alternative, clinical simulation is emerging as a valuable approach to assess DHTs, as it offers a quick, safe, and cost-effective way to test them before implementation [38]. Complementing traditional approaches, this method provides strong evidence through near-real clinical scenarios while allowing for timely and cost-effective updates [38].

A systematic review and meta-analysis focusing on behaviour change apps using RCTs to evaluate mHealth apps reported that most studies conducted parallel or cluster designs, with 34% characterised as pilot studies [65]. Notably, in 83% of cases, control groups did not receive any intervention, and follow-up periods were predominantly limited to six months or less. Additionally, 94% of all studies were conducted in high-income countries, posing challenges in generalising the findings to LMICs where limited human resources and access to healthcare services may present significant challenges [65]. In the same vein, a review of six systematic reviews focused on RCTs evaluating mHealth apps reported that most trials were pilot studies testing feasibility on small populations for short durations [66]. Moreover, many RCTs did not adhere to the relevant reporting guidelines (e.g., CONSORT), less than half demonstrated a meaningful effect on health outcomes, and the overall quality of evidence was very low, challenging their recommendation for use [66].

A scoping review that explored DHT evaluation methods beyond RCTs identified alternative designs such as micro randomisation trials (MRT), fractional RCTs, sequential multiple assignment RTs (SMART), and stepped-wedge cluster RTs [67]. However, the study noted a limited use of these alternative designs, accounting for only 25% of the assessed studies, attributed to challenges in securing funding or a lack of awareness about pre-registering studies using such designs [67].

One study evaluated the characteristics of studies registered in the clinicalTrials.gov database, focusing on mobile-, web-, electronic-based tools, and digital medical devices [69]. The results indicated that 88% of the research designs were interventional, with 71% employing randomisation. Most interventional studies were open-label, and the most frequent approach was parallel assignment. Observational studies predominantly followed a prospective design, with cohort studies being the most common, followed by case-only and case–control approaches.

A scoping review examining the use of DHTs to measure real-world clinical outcomes via patient-generated data revealed that mobile apps constituted the most frequently studied type of DHT (49%) [70]. Among the collected data types, physiological data (37.1%), clinical symptoms data (36.9%), and behavioural data (33.5%) were most prevalent. Furthermore, 75% of the studies leveraged the collected data for descriptive analytical purposes, while predictive, diagnostic, and prescriptive applications were less common (13.7%, 6.5%, and 5.3%, respectively). Additionally, only 12% of the apps incorporated AI for data collection or measurement [70].

Finally, a systematic scoping review of eHealth evaluation methodologies identified 50 unique approaches [68]. From these, the authors formulated a comprehensive lifecycle evaluation framework encompassing six distinct phases (conceptual and planning, design, development and usability, pilot [feasibility], effectiveness [impact], uptake or implementation, and all phases), offering a methodology guide for practical implementation. The chosen approach in each phase can be based on a lifecycle model or an existing framework, depending on the specific DHT under consideration [68]. The guide assigns 75 specific evaluation methods, study designs, frameworks, and evaluation approaches to their respective study phases, accessible to evaluators through a web-based resource [68].

## 4. Discussion

The purpose of this study was to explore the state of the art in DHT evaluation through an analysis of published literature. The bibliometric analysis highlighted a significant rise in publication output, particularly from Europe and the United States. The literature review identified important evaluation dimensions and challenges in evaluating DHTs, such as defining evaluation criteria and navigating regulatory frameworks. Additionally, the study explored existing evaluation frameworks, highlighting their diversity and lack of practical guidance. These findings underscore the pressing need for standardised, interdisciplinary approaches to ensure the efficacy, safety, and ethical integrity of DHTs in healthcare.

The results of the bibliometric analysis indicated a rising interest and focus on DHTs in the academic and research community and suggested that it is gaining recognition in high-impact journals. The highest growth was observed from 2019 to 2020, which is in agreement with previous bibliometric analyses and has been attributed to the recent COVID-19 pandemic, which compelled healthcare organisations and professionals to adopt digital technologies in response to the evolving healthcare landscape [71,72,73]. The distribution across journal categories, with a significant presence in “health care sciences”, “medical informatics”, and “public, environmental, and occupational health”, reflects the interdisciplinary nature of digital health studies. Also, in line with other bibliometric analyses, most of the research related to digital technologies in healthcare is undertaken in developed countries, mainly the UK, the USA, and Australia [71,72]. Besides “eHealth”, “mobile health”, “mobile phone”, and “validation” as primary focus areas within the field, the keyword analysis also showed terms such as “machine learning” and “artificial intelligence”, which can be considered as emerging research hotspots in digital healthcare so far [71]. Regarding DHT evaluation methodologies, the prevalent use of observational studies suggests a significant emphasis on real-world observations and underscores the commitment to rigorous experimental research. Moreover, clustering keywords into different domains showed that the key investigation areas were usability/satisfaction, security/privacy, and feasibility. The comparatively lower priority given to effectiveness, efficacy, and cost–utility could be due to various factors, such as a focus on addressing immediate usability concerns, the exploratory nature of these emerging technologies, or the challenges associated with evaluating the clinical and cost-effectiveness of DHTs.

The challenges identified in the evaluation of DHTs span from the precise definition of DHTs to the rigorous clinical validation process and the establishment of standardised assessment frameworks. Despite efforts by international organisations to provide guidance and standards for DHT evaluation, regulatory frameworks often serve different purposes, such as facilitating market access or determining coverage and reimbursement policies. This divergence creates a complex environment where evaluation criteria and priorities may differ significantly, posing a challenge in establishing a cohesive global standard. Adding to this complexity, accreditation procedures for DHTs show substantial variations from one country to another. This lack of harmonisation results in a notable absence of clarity in evaluation protocols, hindering the development of a universally accepted set of criteria for assessing the performance, safety, and effectiveness of DHTs. Consequently, the absence of a standardised framework can make it challenging for healthcare providers, policymakers, and users to make informed decisions about the selection and integration of DHTs into healthcare systems.

Beyond international bodies, a multitude of DHT evaluation frameworks exist. However, a prevalent challenge is the lack of practical application guidance within these frameworks, presenting obstacles for developers, researchers, and policymakers. Another notable challenge stems from the diverse evaluation approaches and the limited interaction between different perspectives, such as continuous/iterative *versus* lifecycle considerations. This dynamic contributes to uncertainty about the specific aspects to assess at each phase of DHT evaluation, posing practical challenges for stakeholders. Additionally, the variability in evaluation domains, quality criteria, and definitions used across frameworks raises concerns about standardisation and generalizability. A critical issue exacerbating these challenges is the omission of essential criteria in many existing frameworks, resulting in incomplete assessments that focus on specific issues rather than providing a comprehensive understanding.

The infrequent inclusion of clinical effectiveness supported by empirical evidence as a criterion for app quality is of particular significance. This omission complicates the ability to gauge the true impact of DHTs on patient care and overall healthcare outcomes. Furthermore, it is common to observe a misalignment between the medical claims made by Digital Health companies and mHealth apps and the clinical evidence supporting these claims. The disconnect emphasises the importance of ensuring that medical claims are supported by substantial and credible scientific evidence to protect the user’s interests and promote accountability within the industry. In examining the methodology employed to assess clinical effectiveness, it becomes evident from the reviewed literature that RCTs, widely regarded as the gold standard for evaluating healthcare interventions, encounter challenges when applied to the context of DHTs. The findings suggest that the rapidly evolving landscape of digital health necessitates a more adaptive and flexible approach to capture the intricacies and dynamic nature of these technologies. The limitations identified in RCTs, such as their rigidity and often lengthy timelines, highlight the need for alternative methodologies that better align with the pace of innovation in the DHT field. The observed prevalent reliance on prospective studies, particularly pre-post designs, indicates a pragmatic approach to DHT evaluation. This strategy allows for real-world assessments, considering the practical implications and usability of these technologies in diverse healthcare settings. However, prospective studies may have inherent limitations, particularly in capturing long-term outcomes and assessing the broader, systemic impacts of DHTs on healthcare delivery, patient outcomes, and public health. These observations underscore the need for diverse methodologies in evaluating the clinical effectiveness of DHTs, including approaches such as clinical simulation and alternative trial designs. Despite this recognition, the adoption of alternative designs within the DHT field remains limited. This has been attributed to challenges in securing funding and a lack of awareness within the research community, potentially hindering the exploration and application of innovative trial designs tailored to the unique characteristics and requirements of DHTs.

The adoption and integration of DHTs into clinical practice and healthcare systems face barriers beyond the previously mentioned limitations. Firstly, the limited research capacity within the healthcare system hinders the generation of robust evidence, necessitating substantial investments in human and infrastructural resources to conduct rigorous research studies, including clinical trials and evaluations. Secondly, the development, implementation, and maintenance of DHTs require substantial financial investments, posing challenges for healthcare organisations, especially those in resource-constrained environments, in allocating budgets for adopting and integrating DHTs, impacting the pace and scale of implementation. Thirdly, the sensitive nature of health data raises legitimate concerns about data security and potential misuse, requiring robust cybersecurity measures, clear regulations, and transparent communication to build trust among stakeholders. Fourthly, the successful integration of DHTs depends on the understanding and acceptance of healthcare professionals. Insufficient education and awareness programs for clinicians on DHT functionalities, benefits, and the best practices may be contributing to the resistance to change and impede effective incorporation into their practice. Fifthly, end-users, including patients and caregivers, often encounter challenges in navigating and utilising DHTs, needing adequate training, support, and guidance to enhance the usability and acceptance of DHTs. In this line, the EU is actively promoting digital skills through the EU Digital Competence Framework for Citizens, aiming to enhance digital literacy across age groups [74]. This framework serves as a reference guide for individuals, educators, and policymakers, aiding in understanding and developing digital competencies. Lastly, despite the abundance of health apps and digital tools, systematic approaches for disseminating appropriate interventions are lacking. Indeed, the diverse landscape of DHTs, each with unique functionalities, impacts, and costs, contributes to the challenge of ensuring that the right interventions reach the right patients and healthcare providers at the right time.

### 4.1. New Opportunities

Academic-industry collaborations (AICs) have been historically recognised as powerful drivers of innovation, benefiting both sectors through shared resources and expertise [13,75,76]. The establishment of the FDA’s Digital Health Center of Excellence (DHCoE) in 2020 exemplifies a significant initiative supporting digital health stakeholders. This centre provides crucial services, including technological and policy advice, collaboration facilitation, and consistent evaluation quality assurance. Furthermore, a growing trend among academic centres involves creating internal digital innovation units. Examples include the Center for Digital Innovation at the University of California San Francisco, the Center for Digital Health at Stanford Medicine, and the Center for Digital Health & Data Science at Thomas Jefferson University. Additionally, university teaching hospitals and academic health centres are establishing “digital health labs”, such as the Digital Innovation Hub (DHIG) at Brigham and Women’s Hospital (BWH) in Melbourne [77], the Digital Health Translation and Implementation Program (DHTI) at Murdoch Children’s Research Institute (MCRI) [78], and the Digital Health Validation Lab at the University of Glasgow in Scotland. These labs function as collaborative ecosystems, providing a secure platform for industry players, innovators, and start-ups to access clinical resources and data for testing and validating digital health innovations.

#### The Digital Health Validation Center

While academic research provides valuable insights, much of it tends to remain at a theoretical level, sometimes detached from the practical realities of healthcare. In contrast, healthcare centres, such as tertiary hospitals equipped with advanced knowledge and expertise, are uniquely positioned to assume the role of validating centres. In this context, the Digital Health Validation Center, located in the Sant Pau Campus Salut Barcelona, was founded in 2022 as a joint initiative between the Hospital de la Santa Creu i Sant Pau, and the Barcelona Health Hub. This collaboration extends beyond academia, involving robust partnerships between public healthcare centres and private entities across various sectors, including the technology industry, consulting firms in the healthcare and innovation industry, scientific societies, and pharmaceutical companies, among several other health or scientific institutions. These partnerships represent a multifaceted approach to addressing healthcare challenges, leveraging the expertise and resources of both public and private sectors.

Addressing the need for robust validation processes to ensure the efficacy and reliability of DHTs, given their increasing integration into daily healthcare practices, the Digital Health Validation Center has two main objectives. Firstly, to generate clinical and scientific evidence on the impact of digital health. Secondly, to facilitate the technology transference of viable, user-friendly, safe and efficient DHTs from the industry to health and social care providers. To achieve these objectives, the centre operates within a well-structured framework involving various entities (Figure 4), namely a Scientific Coordinating Committee composed by a multidisciplinary team specialised in digital health, an innovation subcommittee of the Sant Pau’s Ethics Committee for Investigation specialised in health Technology (CEIT), an international advisory scientific committee for the design of the validation study of each DHT evaluated, the Sant Pau Hospital Research Institute, and an international network of healthcare centres.

The Digital Health Validation Center provides tailored clinical-technological advice to its collaborators like the co-creation and proof of concept of innovative health technologies, the optimisation of healthcare processes and care routes with the support of the technology, and the customisation of clinical protocols for testing, piloting, or validation. It also functions as a proficient Clinical Research Organization (CRO), employing agile methodologies that reinforce validation efficiency and quality. Furthermore, the Digital Health Validation Center conducts comprehensive analyses encompassing clinical, methodological, technological, statistics, ethical and legal dimensions for ensuring adherence to regulatory standards and certifications. This comprehensive analysis considers not only the patient’s perspective but also the high diversity level and gender perspective. Moreover, the Digital Health Validation Center has the capability to establish connections with other healthcare centres for multicentric validations, thereby broadening the scope of validations at an international level. Through global networking, the centre allows the inclusion of clinical experts from different clinical settings and a larger sample of patients in the studies.

Given the challenges identified in the literature review (such as complex validation processes, regulatory complexities of DHTs regarding ethics, data protection and security, the diversity of evaluation frameworks, the increasing focus on user-centric approaches including accessibility, digital literacy, other socio-economic aspects, or complexity in adoption), the Digital Health Validation Center proposes a scientific and methodological approach to address these issues effectively.

Firstly, the Digital Health Validation Center’s differential factor relies in its internal structure and network of stakeholders, including an innovation hub (BHH), public health sector, and industrial innovation sector. The internal structure, the Scientific Coordinating Committee, is composed of healthcare professionals, bioengineers and telecommunication engineers, information system (IT) experts, epidemiologists, statisticians, medical writers, study coordinators, experts on ethics, legal and regulatory issues, patient advocates, or diffusion and brand experts, among other relevant roles. This structure is complemented by an advisory committee, including several experts in the fields of technology transfer, research or strategy (https://ehealthvalidationcenter.santpau.cat/en/web/public/qui-som (accessed on 10 May 2024)). This collaborative effort expedites contractual and regulation aspects of clinical validation design and execution, ensuring efficient design and execution while guaranteeing the allocation of necessary resources for successful implementation.

Secondly, it ensures that all relevant aspects of health technologies are covered. Based on the existing international framework Guidance and Impact Tracking System (GAITS), an initial assessment of the maturity of a given DHTs is performed to tailor services (Figure 5). This assessment guides the tailoring of services, encompassing co-creation and design for initial stages of maturity; proof-of-concept of DHT for initial tastes, often related to viability, usability or acceptability; and clinical validation through clinical trials including a larger number of participants and specific clinical endpoints.

Thirdly, the Digital Health Validation Center adopts a user-centric approach, incorporating a patient advocate role in every activity among other users depending on the project needs, as well as with experts to incorporate the perspective of the functional diversity of people and the gender perspective. Finally, the Center works towards adoption, including implementation aspects such as integration, change management, and business model considerations, into the validation activities. This integration aligns with the expected outcomes of the services provided, ensuring a holistic and impactful approach throughout the validation process. Thus, the generation of scientific evidence from the validations involves the evaluation of different health outcomes of digital solutions that could be included in the following areas: technological, humanistic, clinical and healthcare, business and legal, all of them in a transversal framework of preservation of ethics in health.

The Digital Health Validation Center is based on more than one decade of experience in creating, validating, and integrating technology into healthcare pathways. Thanks to the Sant Pau Campus Salut Barcelona’s support, the Center for Clinical Validation of Digital Solutions is overseeing 39 digital health projects, fostering cultural change through interdisciplinary collaboration with various stakeholders. Out of these projects, 27 are trials that are currently being designed, following patients, or have been carried out. These projects include telemedicine programs for pathology control, telemonitoring for conditions like cardiac transplants and neurological diseases, and innovative technologies such as virtual reality for invasive procedures. Other initiatives involve portable biomarkers for dermatological diseases, telerehabilitation for pulmonary diseases, AI-powered software for appointment management, and virtual reality therapy for mental health conditions like schizophrenia.

To sum up, the Digital Health Validation Center aims to ensure fast, efficient, appropriate, user-centric, and implementation-oriented services, contribute to scientific evidence on the impact of DHTs, and facilitate public–private collaborations between the healthcare sector and the DHT industry.

### 4.2. Limitations

This study has some limitations that must be acknowledged. Firstly, this study analysed publications from just one database (PubMed). Secondly, the MeSH thesaurus was not extensively used in the PubMed search due to the lack of well-defined terms in the eHealth, mHealth, and digital health fields. Instead, Other Terms (OT), which are keywords selected by authors, were preferred. This choice may introduce potential distortions in search results, deviating from a standardised thesaurus-based approach and reflecting the subjective nature of authors in defining descriptive keywords. Additionally, it is important to note that this review focused on peer-reviewed scientific papers, potentially limiting the inclusion of relevant studies not indexed in PubMed, those written in languages other than English, and grey literature. Since the adoption of DHTs is scarce, most of the initiatives may not have published papers since adoption falls more on the public administration and healthcare field rather than in clinical research areas. Also, most initiatives for adoption are promoted by regions or healthcare providers, with no scientific approach to validation. These limitations could have impacted the identification of additional insights. Moreover, the bibliometric analysis was based on the automatic word search, which entails certain intrinsic limitations that must be considered in the interpretation of the results. These limitations include a lack of context, language evolution, synonyms and linguistic variations that might use different words or phrases to describe similar concepts. Consequently, there is a potential risk of under- or over-representation of the identified dimensions and designs.

## 5. Conclusions

The findings emphasise the need for harmonising evaluation approaches, placing increased emphasis on regulatory clarity, and making a collective effort to integrate robust criteria and empirical evidence across the entire lifecycle of DHTs. Moreover, it is imperative to address existing challenges related to accreditation, adoption, and integration to ensure the quality, efficacy, and safety of DHTs within diverse healthcare settings. Lastly, fostering an environment that promotes innovation, acknowledges regional variations, and prioritises patient-centric approaches is essential for realising the full potential of DHTs and improving overall healthcare delivery. The Digital Health Validation Center is aligned with this purpose, as it seeks to contribute to the scientific evidence on the impact of DHTs from inter-collaborative studies between the healthcare sector and the DHT industry to address the challenges identified.

## Figures and Tables

**Figure 1 healthcare-12-01057-f001:**
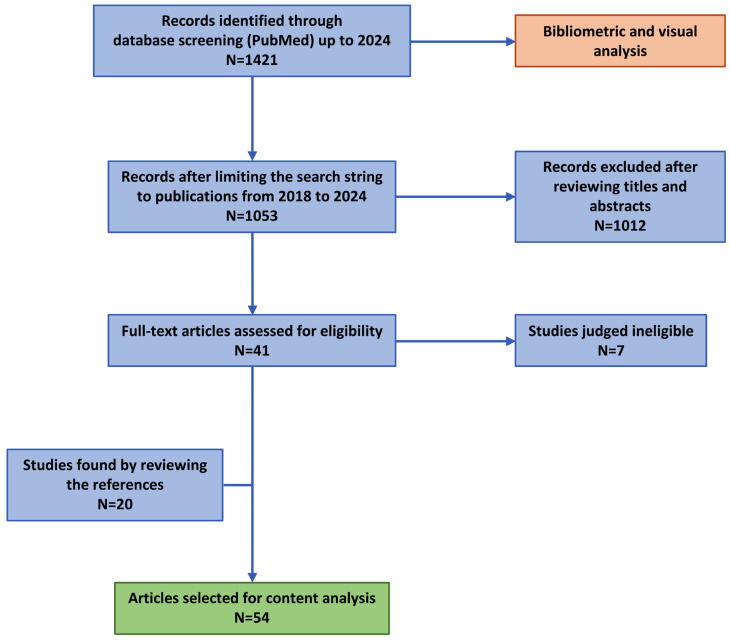
Study selection flow diagram. The figure illustrates the process by which articles were selected for inclusion in the study.

**Figure 2 healthcare-12-01057-f002:**
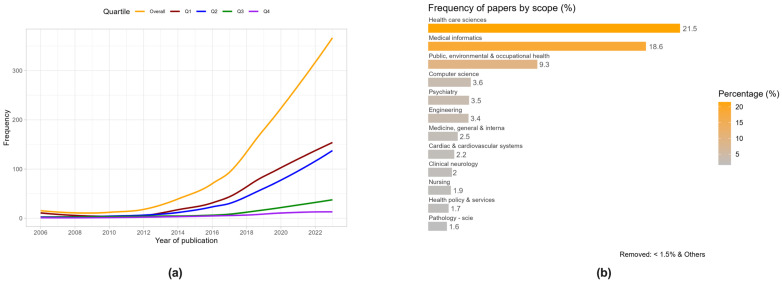
Results of the bibliometric analysis showing (**a**) the evolution of published papers indexed in PubMed by quartile and (**b**) the frequency of papers by scope.

**Figure 3 healthcare-12-01057-f003:**
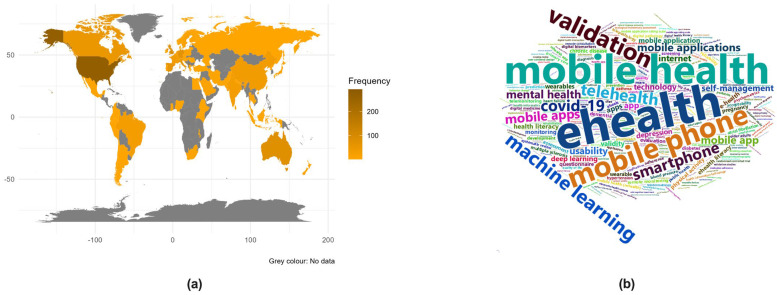
Results of the visual analysis showing (**a**) the map of the frequency of papers by countries worldwide and (**b**) the word cloud of the more prominent keywords among all papers analysed.

**Figure 4 healthcare-12-01057-f004:**
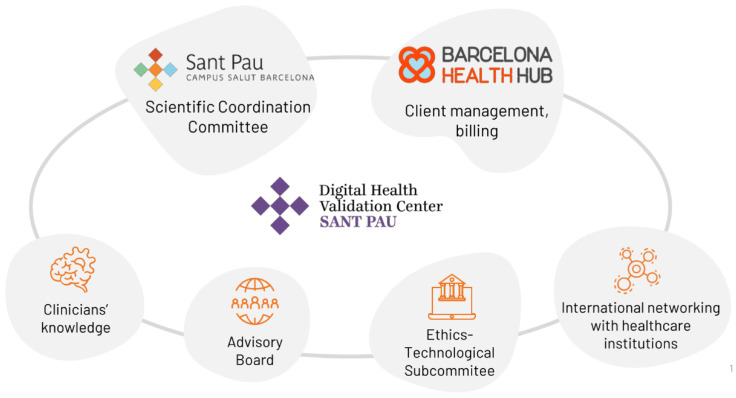
Structure of the Digital Health Validation Center involving various entities, coordinated by the Scientific Coordinating Committee composed of a multidisciplinary team specialised in digital health and research methodology.

**Figure 5 healthcare-12-01057-f005:**
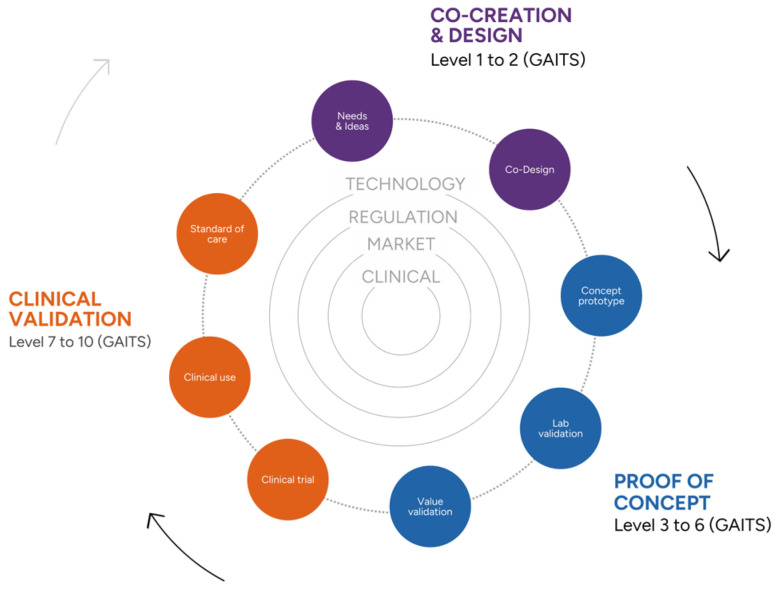
Graphic showing an adapted version of a Guidance and Impact Tracking System (GAITS) healthcare innovation cycle for a digital medicine solution at an early stage at risk due to excessive focus on technology and poor attention to aspects of the clinical domain. Purple: level 1–2, Initial idea and Co-Design; Blue: Proof of concept; Orange: Clinical validation. The Digital Health Validation Center provides support in 4 areas of DHT: Technology, regulation, market and clinical. The services offered includes co-creation and design for initial stages of maturity; proof-of-concept of DHT for initial tastes, and clinical validation through clinical trials including a larger number of participants and specific clinical endpoints.

## Data Availability

Data supporting reported results can be found in https://github.com/CVCSD/Clinical-Validation-StateOfArt/tree/main/data (accessed on 10 May 2024) and https://github.com/CVCSD/Clinical-Validation-StateOfArt (accessed on 10 May 2024).

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
