# Peer review of "Clinical Validation of Digital Healthcare Solutions: State of the Art, Challenges and Opportunities"

_healthcare, 2024, doi:10.3390/healthcare12111057_

Round 1

Reviewer 1 Report

Comments and Suggestions for Authors

The manuscript provides an overview of the development of Digital Health Technologies (DHTs) and the challenges they face, emphasizing the importance of ensuring the effectiveness and safety of DHTs. It advocates the need to harmonize evaluation methods and frameworks, enhance regulatory transparency, and commit to collaboration to integrate rigorous clinical validation and evidence throughout the entire lifecycle of DHTs. 

Some suggestions:

  1. The manuscript lacks coverage of the latest research, as the literature discussed in the text is only up to July 27, 2023. It has been eight months since then, and the authors are advised to add a discussion of recent developments.
  2. All the illustrations in the manuscript become blurry and lose detail when enlarged. The authors are suggested to improve the clarity of the figures or replace them with vector images.
  3. In the discussion section of the manuscript, the section titles and numbering are mixed up, resulting in poorer readability. There is no subsection 4.1, and subsection 4.2 is immediately followed by subsection 4.3.1.
  4. Some parts of the manuscript show shading that needs to be removed. For example, the word "Overall," on line 277.
  5. In Figure 1, the capitalization of the letter 'N' is not consistent, and the boldness of the text in the image is not uniform.
  6. The reference to Figure 1 in lines 126-128 is not appropriate.
  7. The description from lines 145-156 would be more intuitive and readable if accompanied by a legend.

Reviewer 2 Report

Comments and Suggestions for Authors

Dear Authors,

The topic is amazing and impacts the presence of healthcare and health institutions. During the past few years, AI and Blockchain have been the main topics in my workflow; I hope to contribute positively to the author's performance in DHTs. The content of the present manuscript needs a profound restructuring. I will present my suggestions for improving the manuscript:

In the introduction section, the authors should identify the Digital technologies and clarify the aim of their study.

In the methodology section, the authors should identify the question for research and justify why the exclusion criteria: specific pathologies.

Section 3.2.1.1 highlights challenges as findings of the manuscript; it should be clarified, for example, safety perspectives, individual, and DHT specificity. I suggest presenting the results as a feature and discussing them in the proper section.

The authors state, "Finally, it is important to implement rigorous procedures to assess the safety of new DHTs, and it is essential to incorporate perspectives from low- and middle-income countries (LMIC) where the use of digital tools is on the rise." They should clarify based on the literature review and discuss in the section.

Lines 237-238 should identify the countries with the evaluated protocol as the subsequent information.

Section 3.2.1.4. followed one manuscript reference. Why did the authors highlight this section? The authors did not focus on national regulation on this topic. I suggest the article: Reflections about Blockchain in Health Data Sharing: Navigating a Disruptive Technology. DOI: 10.3390/ijerph21020230.

The authors should evaluate the quality and relevance of review articles and any other source to ensure they contribute meaningfully to the argument or discussion presented in the literature review.

The first paragraph of the discussion section is not intended to summarize the work but rather its contextualization with what has been done (e.g., traditional method). The work focused on research; the academic study needs to be differentiated, raising the discussion without support from previous results. The topic may intersect common interests, but this work has other focuses. The initial paragraphs of the discussion serve as a basis for identifying flaws that support the presentation of a validation tool - section 4.3.1. The authors present their opinion on RCTs, going so far as to state that they encounter challenges in the DHT context, which requires a vital discussion. Do the authors have clinical work experience, or are they just dedicated to conceptual studies? They present an exciting tool in a potential proof of concept design. Figure 4 needs to be more didactic; clouds appear without interconnection and without understanding their relationship. Authors should present their conceptual model according to the traditional proof of concept design, giving it deserved prominence. The text presented in the discussion needs to have critical information on the conceptualization of the tool.

When presenting the limitations, I asked the authors whether presenting a scoping review would have been more interesting.

We must wait for data evidence on the development of the tool (section 4.3.1) and scientific presentation of results so that the authors can substantiate the last paragraph of the conclusion section.

Reviewer 3 Report

Comments and Suggestions for Authors

The followings are my observations and suggestions,

1) This is a review paper, which need to review all the exiting state of the methodologies of the said topic which is not sufficiently research in this paper.

2) Data Collection, methodology description lacks clarity, specially how they collected different papers, what are the software the used is not clearly stated and many vague statements are made.

3) Many figures are collected from the internet without source specification. The permission of use is also not mention in this paper.

4) Paper includes a Github link, but what I observed many inconsistency are there.

5) Paper is very poorly written and lacks novelty. 

Comments on the Quality of English Language

The followings are my observations and suggestions,

1) This is a review paper, which need to review all the exiting state of the methodologies of the said topic which is not sufficiently research in this paper.

2) Data Collection, methodology description lacks clarity, specially how they collected different papers, what are the software the used is not clearly stated and many vague statements are made.

3) Many figures are collected from the internet without source specification. The permission of use is also not mention in this paper.

4) Paper includes a Github link, but what I observed many inconsistency are there.

5) Paper is very poorly written and lacks novelty. 

Round 2

Reviewer 2 Report

Comments and Suggestions for Authors

Dear Authors,

The manuscript is substantially improved! I can highlight two suggestions:

According to the comments identified in responses 4) and 5), the topic in question is constantly changing, and regardless of the search for articles, its discussion can be included in subsequent articles as long as it is relevant to the focus of the manuscript.

I also feel it is necessary to clarify ‘traditional RCTs, considered the gold standard for evaluating healthcare interventions, encounter challenges in the context of DHTs.’ How do the traditional ones differ from the current ones? RCTs are clinical studies with a unique structure, and their execution is adjusted to the concepts at that time.  

In the conclusion section, the manuscript does not focus on validating the DHVC  but presents it as an opportunity to disseminate evidence from inter-collaborative studies between academia and industry. I think that's the conclusion. I don't find any reflections in the manuscript on collaborations between public and private institutions, with very different standards and regulations in the health area. 

Author Response

Please find the responses in the attached document.

Reviewer 3 Report

Comments and Suggestions for Authors

This paper improves a lot due to peer review process. I have no further suggestions as the authors addresses well to most of the comments. It could be accepted.

Comments on the Quality of English Language

This paper improves a lot due to peer review process. I have no further suggestions as the authors addresses well to most of the comments. It could be accepted.

Author Response

We sincerely appreciate the time and effort invested by the reviewer in providing valuable feedback to strengthen our paper.

We are grateful for the reviewer's recognition of the enhancements we made to our paper, which were based on his/her feedback. We are pleased to hear that our responses effectively addressed most of the comments, and we are optimistic that our paper will be accepted for publication. Thank you for acknowledging our efforts to improve the quality of our work.